# Application of Best Available Techniques to Remove Air and Water Pollutants from Textile Dyeing and Finishing in South Korea

Gahee Kim [1], Phil-Goo Kang [1,*], Eunseok Kim [1] and Kyungae Seo [2]

1   Integrated Pollution Prevention and Control Research Team, Natural Environment Research Division, National Institute of Environmental Research (NIER), Gyeongseo-dong, Seo-gu, Incheon 22689, Korea; gahkim@korea.kr (G.K.); jugaru@korea.kr (E.K.)
2   Environmental Inspection and Investigation Team, Hanriver Basin Environmental Office, Misagangbyeonhangang-ro, Hanam-si 12902, Gyeonggi-do, Korea; nnke02@korea.kr
*   Correspondence: philgkang@korea.kr

**Abstract:** The textile industry in South Korea is characterized by excessive water consumption, high concentrations of wastewater, hazardous chemicals, and high energy consumption. This study aimed to analyze Best Available Techniques Reference Documents (BREFs) based on best available techniques (BATs) and BAT-associated emission levels (BAT-AELs) and identify potential solutions for tackling environmental pressure from the South Korean textile industry. Therefore, the existing practices of the textile dyeing and finishing industry in South Korea were compared with those from the BREFs of the European Union. Many existing BATs in South Korea are related to reducing water consumption. There is also a strong focus on BATs for reducing wastewater discharge and achieving energy-saving during treatment rather than after treatment, which differs from other industries. Moreover, BAT-AELs were derived for chemical oxygen demand, suspended solids, and total nitrogen for treating non-biodegradable, highly polluted wastewater. Furthermore, BREFs related to atmospheric pollution included dust generated from the heated fabrics in the finishing process that contained cadmium and phenolic hydrogen chloride from dyes and raw materials in the fabrics. Notably, the European Union has not specified BAT-AELs for the textile industry, whereas South Korea has tailored BAT-AELs for toxic and hazardous chemicals. Thus, numerous green techniques to reduce emissions and energy consumption are being implemented in South Korea.

**Keywords:** BAT; BAT-AEL; K-BREF; environmental integrated permit; textile industry; South Korea

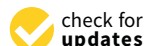



## 1. Introduction

The Integrated Environmental Permit System was introduced in the European Union (EU) in 1996 as an integrated environmental management system for pollution prevention and control in different industries. Subsequently, the EU published Best Available Techniques Reference Documents (BREFs), which specify techniques for controlling pollutants, pollution emission levels, and other information, to support this system [1,2]. South Korea enacted the Act on the Integrated Control of Pollutant-Discharging Facilities on 1 January 2017, which applies to water and air quality for Type 1 and Type 2 business sites across 19 industries. Type 1 sites include workplaces that generate >80 tons of air pollutants per year or emit >2000 m$^3$ of water pollutants per day. In contrast, Type 2 sites include workplaces that generate 20–80 tons of air pollutants per year and emit 700–2000 m$^3$ of water pollutants per day [3,4]. The Integrated Environmental Permission System of South Korea applies to each business site (unlike in the EU) because it is focused on large-scale business sites. This permit system aims to establish a method that can control discharge facilities in an integrated manner and facilitate the implementation of optimal environmental management techniques. These techniques can be applied to each business site in line with

their features to effectively reduce pollutants generated from their corresponding economic activities [5]. Therefore, a customized license/permission system can be introduced with one integrated permit instead of approximately 10 licenses and permits [6,7]. In particular, the Integrated Environmental Permission System is composed of BREFs, which are required for integrated environmental control planning, emission impact analysis, and an integrated environmental control system [8,9]. From this perspective, BREFs include the current status and overall processes of representative industries, including the primary industrial knowledge regarding various processes and techniques, ranging from raw material input to pollutant emissions and treatment techniques.

BREFs are normally compiled via a comprehensive examination of business sites included in the Integrated Environmental Permission System, which considers the current status of each industry, emitted pollutants, and the best available techniques (BATs) that are economically achievable. BREFs play the role of providing business sites in the industries subject to integrated environmental control with BATs to reduce the generation and emission of pollutants and support them to efficiently enforce environmental control. In practice, BREFs are used as the reference tool to elucidate the processes when preparing an integrated environmental control plan and evaluate existing techniques and emissions. Furthermore, BAT-associated emission level (BAT-AEL) can be introduced and used as the primary data for quantifying the maximum emission criteria and emission impact analysis. Although BATs are not mandatory, they are becoming practical permission standards for setting the maximum emission criteria. In general, BREFs are published by the BREF Secretariat at the National Institute of Environmental Research (Incheon, South Korea), which provides practical support, such as for the preparation and publication of BREFs. Practical support implies the determination of details, such as the procedure and composition of BREFs, composition, and operation of technical working groups (TWGs), and operation of BREF Deliberation Committee under the Central Environmental Policy Committee [10]. Generally, BREFs are created to establish the standards for an integrated permit, and 17 BREFs have been published for 19 industries. The BREF for the textile dyeing and finishing industry was published in 2020. The domestic textile industry imports, dyes, and processes textiles produced internationally. Unlike other industries, it features multiple materials and fibrous assembly processes. Additionally, the domestic textile industry is mainly composed of small businesses, which use a larger amount of water and discharge high volumes of wastewater in relation to the size of the business. From an environmental perspective, chemicals such as dyes result in high concentrations of hazardous chemicals in discharged water.

The textile industry involves excessive water consumption, high concentrations of wastewater discharge, hazardous chemicals [11], and consumption of large amounts of energy, thereby requiring efficient BREFs to overcome these challenges. The present study aimed to assess the improvement of standards by analyzing the BREFs of South Korea and comparing them with those of the EU. The present study introduced BREFs based on BATs and BAT-AELs for South Korea. The specific objectives of the study were to (1) analyze BREFs of the textile dyeing and finishing industry in South Korea and (2) propose an efficient national management route by comparing the existing practices in South Korea with the BREFs of the EU textile industry.

## 2. Materials and Methods

### 2.1. Processes of the Textile Dyeing and Finishing Industry

The textile industry processes include (1) spinning, which produces yarns; (2) fabric manufacturing, which produces fabrics; (3) dyeing and finishing of the fabricated fabrics, which produce the final clothing. The textile industry is composed of processes with high value-added organic correlations that lead to finished products for clothing, manufacturing, and exports. Notably, the textile industry in South Korea is only composed of the dyeing and finishing industry because all the raw materials are imported from other coun-

tries. The domestic textile industry in South Korea is diminishing as it advances towards international markets.

The textile dyeing and finishing industries focus on processing; they differ considerably from other domestic industries in terms of integrated permits. The textile industry features multiple raw materials, processes, and emitted pollutants, which are changed by the combination and use of materials, fibrous assemblies, and manufacturing processes, as well as by many overlapping and similar processes. Although other industries compose classification systems by-products and processes in the BREF, the textile industry has overlapping processes that are minimized by distinguishing contents by material, fibrous assembly, and process. Thus, all contents of the textile industry can be included when analyzing BATs (Table 1). The BREFs of the textile industry in the EU comprise a wider range of processes than those in South Korea because the former industries grow cotton, produce yarns, and manufacture and process fabrics by themselves.

The textile dyeing and finishing industry inevitably consumes large volumes of water to dye fabrics, thereby generating large amounts of wastewater. In particular, dyeing 1 kg of fabrics requires approximately 200 kg of water. Moreover, the water consumption is large relative to the size of the business [12]. The business size and daily water consumption were compared using the 2019 Water Emission Management System (WEMS) data in this study. To this end, the average daily water consumption of businesses with Type 1 water quality, which are subject to integrated permits, was used.

The comparison of the textile dyeing and finishing industry processes between South Korea and the EU is indicated in Figure 1. Figure 2 shows that the textile industry uses large amounts of water relative to the business size; the textile industry is the third-largest consumer of water, whereas the business size is the smallest, among 17 industries (businesses with 50 employees or less are small according to the Ministry of SMEs and Startups [13]).

**Table 1.** Classification system of the textile dyeing and finishing industry in the South Korean Best Available Techniques Reference Documents (BREFs) [14].

| Classification | Content |
|---|---|
| Materials | Cotton and cellulose<br>Polyester<br>Nylon<br>Wool<br>Silk, etc. |
| Processes | Pre-treatment (singeing, desizing, scouring, mercerizing, bleaching, etc.)<br>Dyeing/Printing<br>Post-treatment (dehydration, drying, coating, finishing) |
| Fibrous assembly types | Raw materials<br>Yarns<br>Weaving<br>Knitting |

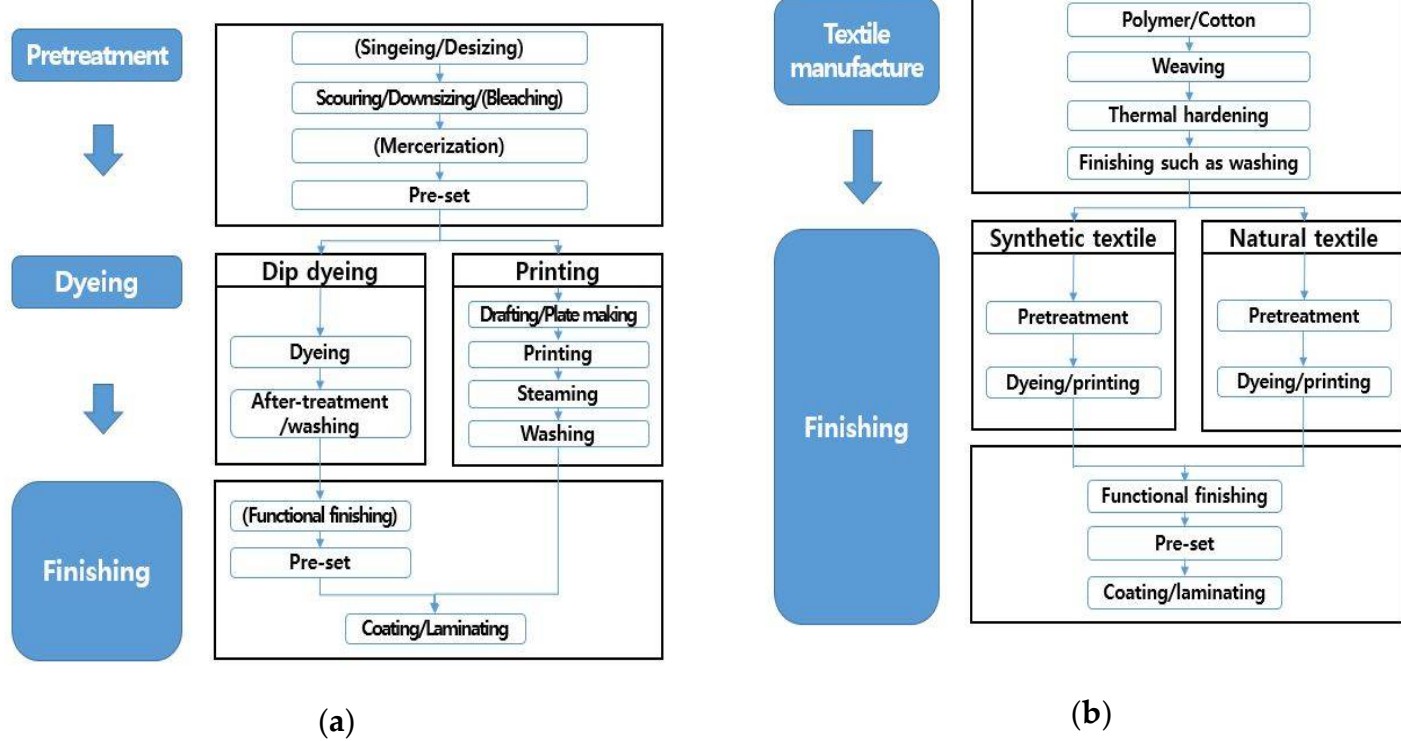

**Figure 1.** Comparison of the textile dyeing and finishing industry processes between (**a**) South Korea and (**b**) the European Union (EU) [14,15].

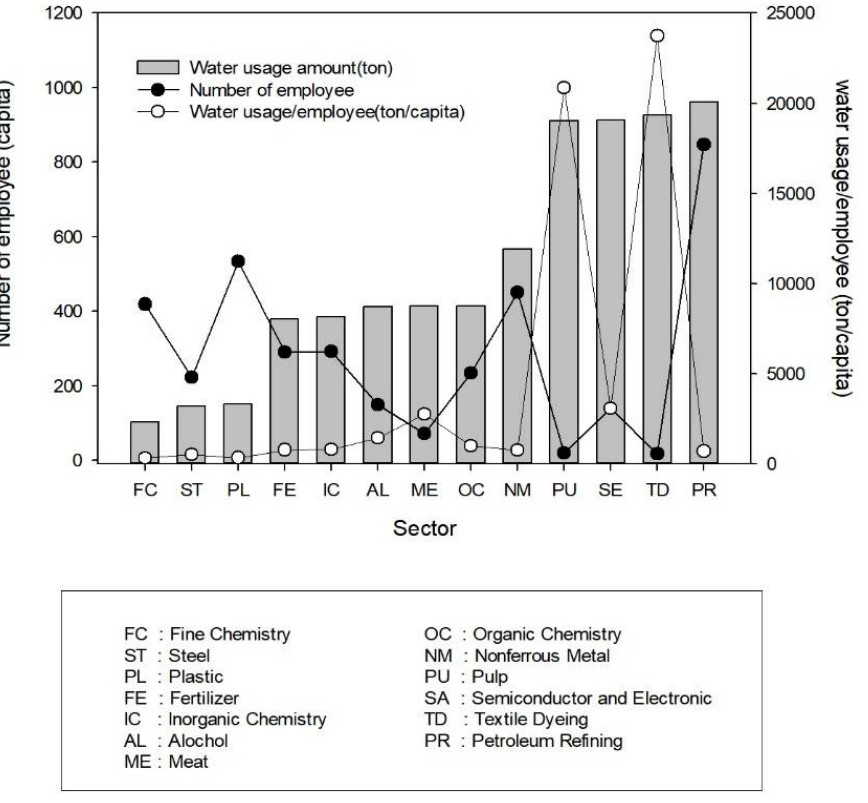

**Figure 2.** Comparison of water consumption by the size of the manufacturing business [16].

### 2.2. Environmental Problems in the Textile Dyeing and Finishing Industry

Owing to the inherent characteristics of the industry, the textile dyeing and finishing industry consumes a large amount of water in the dyeing process and generates high-concentration wastewater because of the use of chemicals such as dyes. Thus, water pollution has emerged as a major environmental issue associated with the textile industry [17]. Wastewater contains high concentrations of hazardous pollutants from dyes and chemicals that are challenging to treat [18]. According to the Ministry of Environment Guidebook for Licensing and Permission of Wastewater Discharge Facilities, 32 types of water pollutants have been identified in the textile dyeing and finishing industry. This list includes 18 types of water pollutants that can exist, such as biochemical oxygen demand(BOD), chemical oxygen demand (COD), and total nitrogen (T-N), and 14 types of specific substances harmful to water quality, such as chrome, cadmium, and formaldehyde. The discharge list is shown in Table 2 [18]. Moreover, large amounts of energy are required to treat industrial wastewater, which is challenging, owing to its high temperature in the industrial treatment systems [19].

The major environmental pollutants discharged from the textile dyeing and finishing industry can be classified into air and water pollutants (see the process-related classification in Tables 2 and 3). All the processes of this industry generate wastewater, as they all use water. Moreover, because chemicals represent the input in the first process, and the following processes are continuously conducted, heavy metals and unreactive dyes are discharged from all processes due to the chemicals in wastewater. Additionally, the COD of the emissions is high. From the perspective of air pollution, dust aerosol (i.e., harmful particulate matter) is mainly generated in the drying process. Moreover, gaseous pollution can be generated by producing formaldehyde from the use of chemicals as raw materials. According to the BREFs of the textile dyeing and finishing industry in the EU, the use of water and chemicals for dyeing and the generation of wastewater are reported as the emerging environmental problems in South Korea. The use of pesticides is also an emerging environmental problem because pesticides are involved in the processes of manufacturing yarns and textiles, such as cotton farming.

**Table 2.** List of water pollutants discharged from the textile dyeing and finishing industry [19].

| Classification | | Pollutants |
|---|---|---|
| Water pollutants discharge list (32 types) | Specific substances harmful to water quality (14 types) | Copper and copper compounds, lead and lead compounds, mercury and mercury compounds, cyanide compounds, hexavalent chrome compounds, cadmium and calcium compounds, dichloromethane, chloroform, 1,4-dioxane, diethyl hexyl phthalate (DEHP), acrylonitrile, naphthalene, formaldehyde, and epichlorohydrin |
| | Water pollutants (18 types, excluding specific substances) | Organic matter (biochemical oxygen demand (BOD) and chemical oxygen demand (COD)), suspended solids (SS), total nitrogen (T-N), total phosphorus (T-P), oils (mineral oil), oils (animal and vegetable oils), nickel and nickel compounds, manganese and manganese compounds, barium compounds, fluorine compounds, detergents, zinc and zinc compounds, iron and iron compounds, chrome and chrome compounds, perchlorate, phenols, and acids and alkalis (pH) |

**Table 3.** Raw materials, process descriptions, and major environmental problems associated with major manufacturing processes in the domestic textile dyeing and finishing industry.

| Process | Raw Materials | Process Description | Major Environmental Problems |
|---|---|---|---|
| Desizing | Weak acid, oxidative acid, water, alkali ($Na_2CO_3$) | A process of adding a chemical (thickener) to impart strength | Air $\rightarrow$ Volatile organic compounds (VOCs) <br> Water $\rightarrow$ High BOD load, discharges many non-biodegradable matters, high concentration of solids |
| Scouring | Alkali (NaOH), surfactant, water | A process of removing impurities for uniform dyeing | Water $\rightarrow$ High pH, BOD, and COD; high alkalinity; and toxic substances |
| Bleaching | Bleach (e.g., $H_2O_2$), water | A process of removing coloring matters after scouring | Water $\rightarrow$ Bleach residues, high pH, SS, and BOD |
| Mercerizing | Alkali (NaOH) | A process of improving the gloss and strength of textiles | Water $\rightarrow$ Strong alkalinity |
| Dyeing | NaOH, dye, surfactant, water, additives (e.g., oxidant) | A process of coloring textiles using chemicals | Air $\rightarrow$ VOCs, energy <br> Water $\rightarrow$ High COD and BOD, odor, alternative oxidase (AOX), heavy metals, additives, discharge of undyed dyes |
| Printing | Pigment, dye, additives (e.g., oxidant) | A process of applying specific patterns or designs to textiles | Air $\rightarrow$ VOCs, methanol, formaldehyde, etc. <br> Water $\rightarrow$ SS, AOX, COD is higher than BOD |
| Finishing | Heat, resin, softener | A process of imparting desired properties to textiles | Air $\rightarrow$ dust, sulfur oxides, formaldehyde, energy, etc. <br> Water $\rightarrow$ SS, energy |

### 2.3. BATs and BAT-AELs of the Textile Dyeing and Finishing Industry

BATs are formulated as a result of the experts' meetings in the corresponding research/industry domain [1,5]. Such expert meetings include TWGs, whose members are primarily the experts at workplaces, industries, and academia. According to Article 24, Paragraph 5 of the Act on the Integrated Control of Pollutant-Discharging Facilities, BREFs are created based on the opinions and decisions of TWG members. They are published after being edited under the supervision of the National Institute of Environmental Research through the deliberation of the Central Environmental Policy Committee. BATs are delivered through the investigation of current status, in-field investigation, and license/permit review for 236 business sites in the textile dyeing and finishing industry subject to the integrated environmental control, primary data survey for three years for Korea, and 20 TWG meetings and small group meetings [14].

Unlike those of other industries, BATs for the textile dyeing and finishing industry have been formulated according to the classification systems that rely mostly on material, fibrous assembly, and process. Thus, the overlapped contents were minimized while describing all three characteristics (material, fibrous assembly, and process). They were classified according to the material within each process, and the shapes of fibrous assemblies were described in the detailed process description of each material. Thus, BATs were largely divided into general BATs, which comprised commonly applied and process BATs. Then, the process BATs were divided into pre-treatment, dyeing, and finishing processes. Additionally, BATs were separately derived according to the material within each process. Moreover, raw materials for each process, pollutants emitted, and BATs for treating pollutants were presented in integrated process diagrams.

BAT-AELs are mainly the lower and upper limits established for a range of pollutant discharge levels during normal operation by applying the derived BATs. Thus, these are the maximum permissible discharge levels that serve as a reference point for the permitted emission standards [20]. The maximum emission criteria are selected based on the pollutant items and upper limits derived from BATs. They are explicitly described in the Act on the Integrated Control of Pollutant-Discharging Facilities and are directly applied to the

business sites. Additionally, the national certified data are used for setting the BAT-AEL values. Therefore, we applied the 2015 data for Stack Emission Management System (SEMS), Tele-Monitoring System (TMS), Water Emission Management System (WEMS), and Water Tele-Monitoring System (WTMS) of the business sites subject to the integrated environmental control of the textile dyeing and finishing industry [14]. The air pollutants were classified by the type of facilities according to the Clean Air Conservation Act. In contrast, the water pollutants were classified by region according to the Water Environment Conservation Act. Subsequently, the classification system for calculating linked emission levels was established, and BAT-AELs were calculated according to Seo et al. [21] by considering the classification system of BREFs.

During this process, COD, SS, and T-N were derived; they are all essential for treatment because wastewater is highly concentrated and contains mostly non-biodegradable components. Furthermore, the total suspended particles generated from heating the fabrics in the finishing process and the cadmium and phenolic hydrogen chloride generated from the dye residues in fabrics were included in the considered air pollutants. In particular, pollutants such as cadmium and phenol are classified as hazardous chemicals, and only trace amounts are discharged from the textile dyeing and finishing industry. It should also be noted that they were additionally selected for BAT-AEL through TWG meetings because they were considered as the subject of priority control, due to being hazardous chemicals that can harm the human body.

## 3. Results and Discussion

### 3.1. BATs

The integrated process diagrams related to the textile dyeing and finishing processes that apply BATs are presented in Figures 3 and 4.

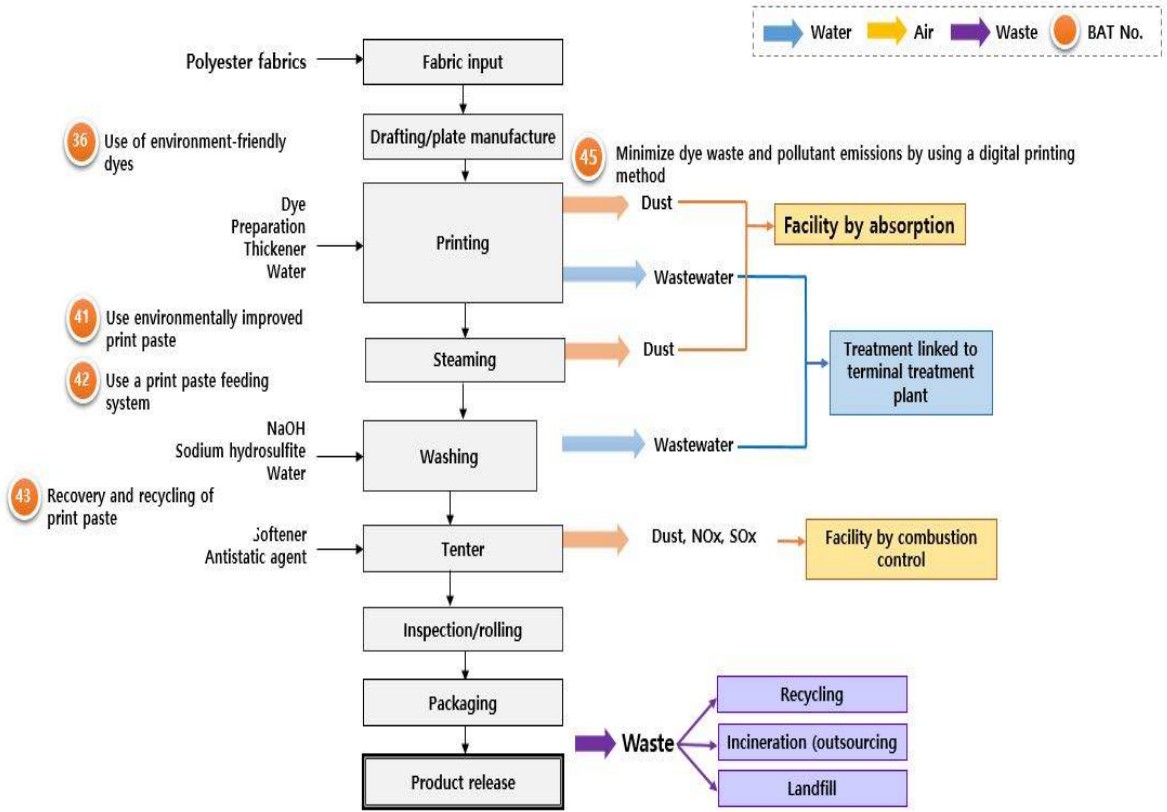

**Figure 3.** Example of integrated process diagram and BATs for the printing process (BAT No. is the number of BREF as shown in Table 4).

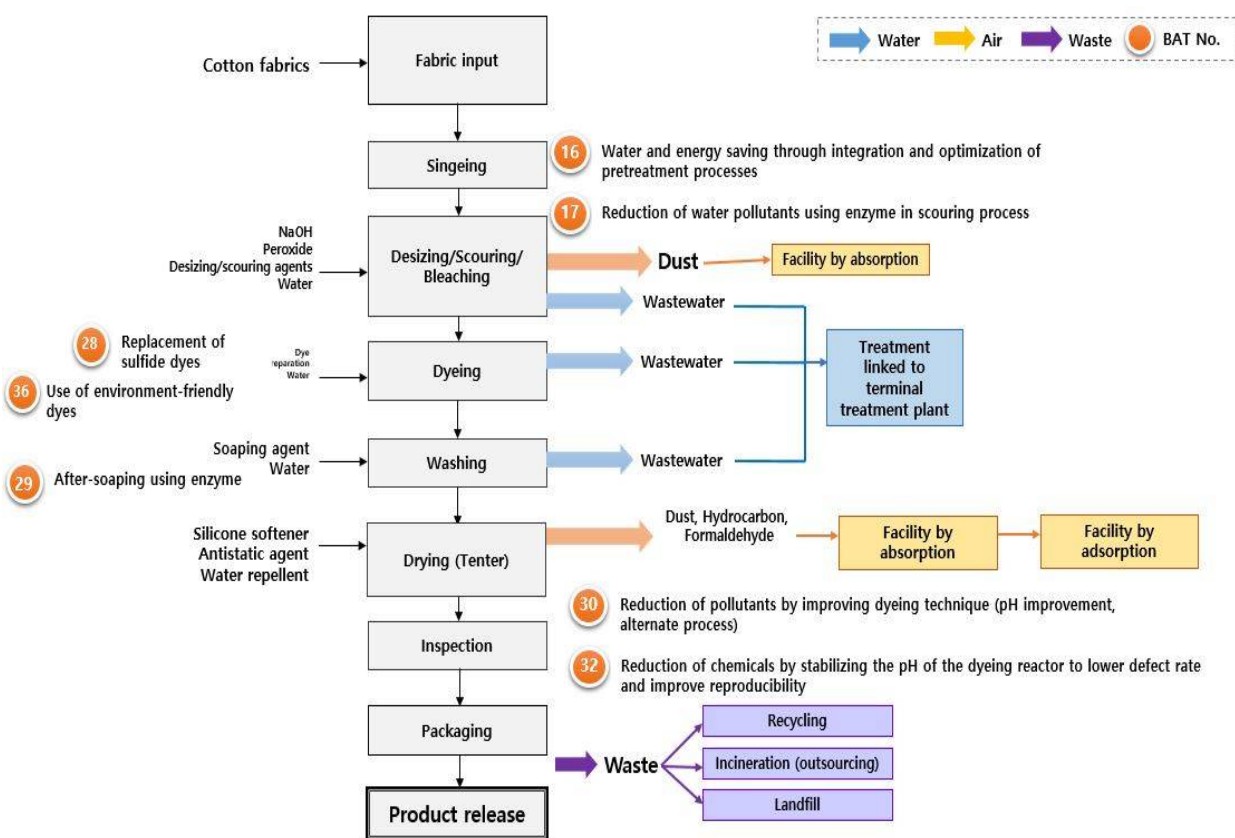

**Figure 4.** Example of integrated process diagram and best available techniques (BATs) for the cotton and cellulose dip dyeing process (BAT No. is the number of BREF as shown in Table 4).

The textile dyeing and finishing industry features numerous BATs for processes to reduce water consumption, considering the discharge of wastewater-centered pollutants and excessive water consumption. For the final versions of BATs, discharge and energy-saving techniques were particularly high in the manufacturing and finishing processes, rather than after treatment, compared with the currently published BATs of other industries (Table 5). In other industries, most BATs are often associated with post-treatment techniques, that is, treatment after discharge. However, the ratio of process BATs in the textile dyeing and finishing industry was found to be 84%, which is nearly twofold higher than that of discharge reduction techniques in processes (47.5%) among the 13 currently published BREFs. Additionally, the BATs for saving water by reusing water or integrating processes were derived and shown in Table 5. This indicates that many green techniques to reduce discharges and energy consumption are being developed and used.

**Table 4.** Examples for pollutant reduction BATs in the production processes of the textile dyeing and finishing industry.

| Classification | BAT No * | Description |
|---|---|---|
| Dip dyeing | 16 | A technique to save water and energy consumption by integrating and optimizing individual pre-treatment processes applied to cotton materials.<br>a. Application of integrated pre-treatment process for cotton textiles<br>b. Optimization of cotton warp pre-treatment |
| | 32 | A technique to decrease the defect rate by stabilizing the pH of the bath in which dyeing reaction occurs and to reduce the use of chemicals by improving reproducibility |
| Printing | 43 | A technique to reduce pollutant discharge through the recovery and recycling of print paste<br>a. Recovery of print paste from the supply system of a rotary screen printing machine<br>b. Recycling of residual print pastes |
| | 45 | A technique to minimize the waste of dyes and the discharge of pollutants using a digital printing method<br>a. Use of a digital jet printing method for bulky textiles<br>b. Use of inkjet digital printing method for flat textiles |

* BAT number of BREF for textile industry [14].

**Table 5.** Comparison of pollutant reduction BAT ratios in production processes.

| Classification | Number of BATs | |
|---|---|---|
| | In Manufacturing Facilities | In Prevention Facilities |
| BATs in textile manufacturing BREF, Korea | 42 (84%) | 8 (12%) |
| BATs in BREFs, Korea | 590 (48%) | 651 (52%) |

### 3.2. Comparison with EU-BREF

Similar to South Korea, the EU has formulated general BATs, which can be applied to the entire textile industry, and process BATs, which are applicable only to certain processes. Similar techniques are included in the general and process BATs. This implies that the businesses of the textile industries in South Korea and the EU are currently using similar techniques. The number of BATs in the EU is 113, which is nearly twofold, compared with BATs in South Korea (Table 6). This difference is driven by the existence of yarn- and fabric-manufacturing processes. Furthermore, in the EU, the processes and BATs of the carpet industry are also specified. Moreover, separate BATs for effluents and wastes are derived among BATs in the EU; for example, BAT for monitoring effluents, wherein the discharged cooling water is reused as process water, and the solid waste is collected and disposed of separately [15]. Unlike South Korea, the EU does not use many process techniques, while the wastewater of the textile industry is generally non-biodegradable. Notably, wastewater treatment is important because of the high concentration of organic matter in the wastewater. In particular, the dedicated BATs for wastewater must be highlighted for this purpose. Among the conventional BATs, those associated with improving the water and energy consumption efficiencies by integrating and optimizing individual processes in pre-treatment can be used for this purpose. There were common BATs for using automated systems in dye injection and supply and recycling water used in the dyeing process. In the printing process, common BATs included the techniques for minimizing dye wastes and pollutant discharges by utilizing a digital printing method and advanced techniques such as environmentally friendly printing thickener. In the finishing process, as in the printing process, many common BATs for saving energy have been formulated, such as heat recovery or smart inverter installation when operating tenter (dryer) facilities.

For example, in South Korea, there are wastewater-related BATs to reduce the discharge of water pollutants using the enzymes in the scouring process for pre-treatment and reduce pollutant discharge, such as minimizing the use of chlorine-based bleach. In the EU, there are BATs to facilitate wastewater treatment using either biodegradable lubricant or an easily degradable thickener. From the dyeing perspective, there are BATs in both the EU and South Korea for reducing chemical use by stabilizing pH in dyeing reaction and for reducing chemical discharge by removing non-permeated dyes on the textile surface. In addition, techniques to apply a chrome-free dyeing method for wool dyeing have been derived as a BAT. Specifically, the chrome-free dyeing method has been listed as an environmental control technique in the EU BREF, but it has not been considered as a BAT in South Korea. However, BATs have been proposed for decreasing discharge by the reduced use of chemicals in the dyeing process. The EU has also formulated techniques to reduce chemical use, but they are characterized by many BATs pertaining to the dyeing process for carpet and wool. Notably, these techniques are frequently used in Europe. From the printing perspective, BATs for minimizing dyes have been derived using a digital printing method, thereby reducing pollutant discharge. This is achieved using environmentally friendly print pastes in South Korea and the EU. However, the technique for improving the economy using pigments with a lower production cost than dyes has been proposed as a BAT only in South Korea.

BATs in the EU are focused on wastewater, whereas South Korean BATs are focused on various pollution sources, including air pollution, odor, and noise. The EU BATs have generally adopted primary industrial techniques by investigating and registering the techniques of many countries that belong to the EU. In contrast, BATs in South Korea have adopted the methods directly used in workplaces through a complete survey of workplaces, resulting in more reasonable results because the individual permit system is based on workplaces rather than on pollutants. Therefore, the BATs of South Korea include environmental techniques in general, such as those pertaining to air pollutants, odor, and wastewater. Furthermore, South Korea could derive more subdivided BATs of the textile industry (based on materials) than the EU. This is recommended because South Korea, in contrast to the EU, seemingly only deals with dyeing and finishing processes.

**Table 6.** Comparison of BATs between Korea BREF (K-BREF) and European Union BREF (EU-BREF).

| Classification | K-BREF | EU-BREF |
| --- | --- | --- |
| | 50 BATs in total | 113 BATs in total |
| BATs | - General (15) <br> - Process (35) | - General (39) <br> - Process (63) <br> - Effluent and wastes (11) |
| Characteristics | BATs about the reduction in general pollutant discharges from business sites as well as about water quality | Separate BATs about the carpet industry and about the treatment of effluent and wastes (sludge) |
| | - Reduction of air pollutants <br> - Reduction of noise and vibrations <br> - Reduction of odor | - Sludge treatment of the carpet industry <br> - Effluent and waste treatment |

*3.3. BAT-AEL*

Table 7 shows the classification system for each textile dyeing and processing industry deriving BAT-AEL. The BAT-AEL derived from analysis based on the classification system is shown in Tables 8 and 9.

**Table 7.** Classification system for pollutant emission facilities in textile dyeing and processing industry for BAT-AEL derivation.

| Main Category | Middle Category | Small Category | Detailed Category |
|---|---|---|---|
| Textile dyeing and finishing industry | Process facilities | Pre-treatment facilities | |
| | | Dyeing facilities | |
| | | Finishing facilities | |
| | Common facilities | Solid particle storage facilities | |
| | Incineration facilities | Waste gas Incineration facilities | 200 kg·h$^{-1}$ |
| | | | Less than 2 t·h$^{-1}$ |
| | | Wastewater/waste incineration facilities | 200 kg·h$^{-1}$ |
| | | | Less than 2 t·h$^{-1}$ |

**Table 8.** BAT-AELs for air pollutants.

| Classification | | BAT-AELs for Air Pollutants | | |
|---|---|---|---|---|
| | | Pollutants | Unit | BAT-AEL |
| Pretreatment | | Dust | mg·Sm$^{3-1}$ | 4–24 |
| | | HCl | ppm | 1–4 |
| | | Cadmium | mg·Sm$^{3-1}$ | 0–0.1 |
| Dyeing/finishing | | HCl | ppm | 1–2 |
| | | Phenol | ppm | 0–2 |
| | | Hydrocarbon | ppm | 7–34 |
| Incineration facilities | Waste gas incineration | Sulfur oxides | mg·Sm$^{3-1}$ | 4–27 |
| | | Carbon monoxide | ppm | 21–154 |
| | | HCl | ppm | 2–13 |
| | Wastewater/waste incineration | Sulfur oxides | mg·Sm$^{3-1}$ | 5–27 |

**Table 9.** BAT-AELs for water pollutants.

| Classification | BAT-AELs for Water Pollutants | | |
|---|---|---|---|
| | Pollutants | Unit | BAT-AEL |
| ≥2000 t/day—clean area | COD | mg·L$^{-1}$ | 18–30 |
| ≥2000 t/day—clean area | SS | mg·L$^{-1}$ | 2–17 |
| Clean area | T-N | mg·L$^{-1}$ | 12–30 |

*3.4. Limitations and Further Studies*

3.4.1. BATs

In South Korea, all environmental management techniques applied in business sites have been considered BATs. Compared with the EU BATs, the South Korean BATs are advantageous because they are site-specific and directly applied. This is because these BATs had been developed through field surveys of all business sites, thereby providing and supporting tailored measures for business sites.

Moreover, although the EU BATs are focused on techniques related to wastewater in the textile industry, the BATs of South Korea have been derived for environmental management techniques. They are tailored to deal with air pollution, odor, and waste managed by business sites, although techniques related to wastewater also exist. However, BATs in South Korea have considerable limitations. As some techniques used by business sites were derived as BATs, the evaluation of economic feasibility and efficiency was insufficient. Moreover, it is not obligatory for business sites to measure economic feasibility and efficiency; therefore, only the design values can be applied. This considerably compromises the economic feasibility analysis of BATs. In contrast, the EU has considered the economic feasibility and environmental benefits of BATs. Thus, South Korean BATs should also be evaluated for economic feasibility and environmental benefits in the future. This can be achieved by analyzing and reviewing the actual data post-management, whereas the annual reports of business sites should be published to reflect the business reality. Additionally, new technologies can be proposed to be used as BATs when BREFs are revised, using more suitable techniques.

### 3.4.2. BAT-AELs

Thus far, BAT-AELs have been derived by the industry only for basic BAT environmental threats: aerosols (dust) and gaseous pollutants (sulfur oxides) in the air and water pollutants such as COD and SS. Additionally, the data used to derive BAT-AELs currently include national certified data such as SEMS, WEMS, TMS, and WTMS. However, they have long measurement and input cycles, and the number of analyzed items remains limited. Therefore, some items cannot be derived due to insufficient sample size in statistical analysis. This problem can be alleviated by collecting the data through annual reports and post-management, as with BATs. From the perspective of water pollution, the analysis of specific substances harmful to water quality is presently mandatory. Therefore, BAT-AELs can be derived further for a broader range of pollutants by utilizing these data.

**Author Contributions:** Conceptualization, G.K., P.-G.K., E.K. and K.S.; investigation, G.K.; resources, G.K. and P.-G.K.; data curation, E.K.; writing—original draft preparation, G.K.; writing—review and editing, G.K. and P.-G.K. All authors have read and agreed to the published version of the manuscript.

**Funding:** This research was funded by the Ministry of Environment, National Institute of Environment Research, Grant Number NIER-2017-01-02-061.

**Acknowledgments:** We would like to express our gratitude to Jaehong Park from the National Institute of Environmental Research, who reviewed this research paper comprehensively, and Joonseok Koh from Konkuk University, who provided support during the field inspection of the fiber dyeing and finishing industries in South Korea.

**Conflicts of Interest:** The authors declare no conflict of interest.

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
