# Peer review of "Application of Best Available Techniques to Remove Air and Water Pollutants from Textile Dyeing and Finishing in South Korea"

_sustainability, doi:10.3390/su14042209_

Round 1

Reviewer 1 Report

Comments:

The author should improve the quality of the figure. 

The authors should be uniformed the units and symbols according to the journal format.

The language contains too many grammar and spelling errors and needs thorough revision.

Reviewer 2 Report

An interesting study, especially important for comparing approaches to the development and application of the best available technologies in the European Union and South Korea.

Reviewer 3 Report

The work is well structured and articulated and can represent an interesting comparative analysis between European and South Korean regulations. The title of the paper however, leads the reader to think that they will find an experimental work where BAT is tested in particular of the textile industry while real case studies are not reduced in the text. The title could therefore be an indicator, making it more consistent with the contents, or insert case studies as examples. Paragraph 3.5.1 also talks about the inconsistency of data for a statistical analysis of elements (lines 346-347), but no statistical result is reported throughout the text. It would therefore be essential to specify this thing better Fig. 1 there are no differences between (a) and (b); is a mistake? In the bibliography, standardize the formatting of the years of publication.
